# Evaluation and Determinants of the Digital Inclusive Financial Support Efficiency for Marine Carbon Sink Fisheries: Evidence from China

**DOI:** 10.3390/ijerph192113971

**Published:** 2022-10-27

**Authors:** Weicheng Xu, Xiangyu Zhu

**Affiliations:** 1School of Economics, Ocean University of China, Qingdao 266100, China; 2Institute of Marine Development, Ocean University of China, Qingdao 266100, China

**Keywords:** marine carbon sink fisheries, digital inclusive finance, grey relational analysis, super-EBM model, grounded theory

## Abstract

The development of digital inclusive finance has greatly improved the feasibility of financial inclusion. Therefore, in the context of the constrained financing of marine carbon sink fisheries, we try to investigate whether digital inclusive finance exhibits a supportive effect on marine carbon sink fisheries and thus enhances the capacity of marine carbon sinks. Specifically, this paper empirically calculates the grey correlation between the development of digital inclusive finance and marine carbon sinks based on data in nine coastal provinces of China from 2011 to 2019. The empirical results show that the grey relational coefficients between the above two in China are more than 0.5, revealing a significant positive correlation. Then, on this basis, we estimate the digital inclusive financial support efficiency (DIFSE) for marine carbon sink fisheries by applying the Super-EBM model. In addition, the determinants affecting the DIFSE for marine carbon sink fisheries selected based on the grounded theory are explored through the Tobit model. The conclusions are as follows. First, there are time-varying characteristics and regional heterogeneity in DIFSE. Generally, the effect of China’s digital inclusive financial support for marine carbon sink fisheries is expanding year by year. Among them, the DIFSE in the northern marine economic circle is currently the highest, followed by that in the south and east. Second, the input of productive factors, promotion of fishery skill, development of fishery technology, and Internet coverage will significantly increase the value of DIFSE, while output structure, income level, fishery disasters, and marine pollution will have significant negative effects on DIFSE. These empirical results can help policymakers better understand the contribution of digital inclusive finance to marine carbon sink fisheries and provide them with valuable information for the formulation of supportive policies.

## 1. Introduction

Since the 21st century, greenhouse gas emissions such as carbon dioxide have increased sharply, exacerbating the global climate problems. Global warming is becoming a serious survival issue facing humanity, and countries around the world are actively exploring effective low-carbon growth models [1,2,3]. In this context, the nineteenth National Congress of the Communist Party of China pointed out that we should endeavor to promote green development, accelerate the reform of the ecological civilization system, address prominent environmental problems, intensify the protection of the ecosystem, and jointly build a beautiful China. According to the statement of President Xi Jinping, China promises to peak carbon dioxide emissions by 2030. Before 2060, greenhouse gases generated by enterprises or individuals will be offset through afforestation, energy conservation, and emissions reduction to achieve carbon neutrality.

With the current development of science and technology, the path of a low-carbon economy mainly includes directly limiting the carbon dioxide emissions of the industry or capturing carbon and using biological carbon sequestration/storage function to implement a biological carbon sink. Compared with mandatory CO_2_ emissions and industrial carbon capture, biological carbon sequestration has the advantages of strong feasibility, low cost, and remarkability. As an important component of biological carbon sequestration, marine carbon sinks can be responsible for up to 40% of the observed decadal variability in atmospheric CO_2_ accumulation [4]. The fishery on marine carbon sink feeding aquatic products with carbon sink functions has great potential in the process of realizing sustainable development and promoting a low-carbon economy [5]. Naturally, the healthy growth of this industry cannot be separated from the strong support of the financial sector. However, there is a widespread problem of asymmetric information in the financial market, and the credit of fishery subjects is relatively weak, which leads to the traditional financial institutions being very cautious in granting loans to fishermen. The financing cost of the fishery is high, the financing scale is limited, and the corresponding financing constraints are also serious [6], slowing down the development of China’s marine carbon sink fisheries and hindering the process of the low-carbon economy.

Digital inclusive finance generally refers to a new financial model developed by traditional financial institutions in cooperation with modern technology Internet companies, using new digital technologies in the financing, payment, investment, and other financial services [7]. Digital inclusive finance is a technology-driven financial innovation, which applies emerging technologies to the financial field. It effectively extends the boundaries of financial services, improves the convenience for financial service demand colony to obtain financial resources, and further strengthens the ability of finance to serve the real economy [8]. In recent years, digital inclusive finance has closely absorbed hundreds of millions of mobile clients with the help of Internet platforms, and the data left by users on the platform are collated and analyzed for the credit assessment to reduce the cost of customer acquisition and risk control, thus greatly improving the feasibility of inclusive finance [9].

Therefore, can digital inclusive finance take advantage of technology-enabling finance to extend financial products and services to the “last mile”, so as to better support the development of China’s marine carbon sink fisheries, thus achieving more carbon sinks? If the results confirm the existence of the support effect, how efficient is this role? What are the characteristics of the spatio-temporal evolution in support efficiency? Finally, what factors can affect this support efficiency? In-depth exploration and discussion of these issues are of great significance for promoting the healthy development of China’s marine carbon sink fisheries and accelerating the construction of China’s low-carbon economy. In particular, regarding the theoretical significance, this paper opens up a new perspective for scholars who wish to conduct research in the area of digital inclusive finance serving the marine economy. In terms of the practical significance, this paper is of importance for policy makers to recognize the role of digital inclusive finance in supporting the development of marine carbon sink fisheries and thus to formulate appropriate policies.

The emergence of digital inclusive finance offers a major opportunity to address the serious piscatorial financing constraints. In this context, for the first time, we explore the role of digital inclusive finance in supporting marine carbon sink fisheries, and we conduct extensive follow-up research. Specifically, this paper empirically examines the correlation between China’s digital inclusive finance level and marine carbon sinks using Deng’s grey relational analysis model based on the data in nine Chinese coastal provinces from 2011 to 2019. On the basis of the existence of a positive correlation between the above two, we measure the digital inclusive financial support efficiency (DIFSE) for marine carbon sink fisheries through the Super-EBM model and analyze its spatio-temporal evolution characteristics. Moreover, we establish an analytical framework of determinants including three dimensions of resource input, economic fundamentals, and environmental impact based on the grounded theory, and the determinants affecting DIFSE are discussed with the help of the Tobit model. Finally, relevant suggestions are put forward to further promote the development of the marine carbon sink fisheries in China.

This paper may contribute in the following aspects. First, we add to the relative literature by exploring empirically, for the first time, the support role of digital inclusive finance on marine carbon sink fisheries. Second, although the data envelopment analysis (DEA) and their extensions have been widely used in financial support efficiency research, the application of EBM models compatible with radial and non-radial data needs to be enhanced. On this basis, this paper measures the DIFSE for marine carbon sink fisheries using the Super-EBM model, which provides a reference for scholars to conduct similar studies in this field. Third, we complement the related literature by comprehensively selecting the determinant indicators affecting DIFSE with the help of the grounded theory for analytical investigation.

The remaining sections of this paper are structured as follows. Section 2 presents the literature review. Section 3 introduces the analytical framework based on grounded theory. Section 4 discusses the sample construction and research design. Section 5 reports the empirical results, and finally, the conclusions are discussed in Section 6.

## 2. Literature Review

### 2.1. Research on Marine Carbon Sink

The carbon sequestration capacity of marine carbon sink organisms is extremely powerful and efficient. Even though their growth area is less than 0.5% of the global seabed area, up to 70% of the carbon is captured by marine plants and is transformed into marine sediments, forming carbon sinks and being stored in the ocean for thousands of years. The participation of marine ecosystems in the carbon cycle is mainly completed by marine biological pumps, and this mechanism can be divided into organic carbon pump and carbonate pump [10]. The organic carbon pump refers to the process by which primary producers such as phytoplankton and algae in the ocean achieve carbon capture through photosynthesis [11], and the carbon-sink mechanism of the calcium carbonate or calcium carbonate debris from marine life transported to the deep sea for burial is the so-called carbonate pump [12].

Gonzalez et al. (2008) measured the carbon sink capacity of marine phytoplankton and found that it captured more than 36.5 Pg (C) of CO_2_ annually through photosynthesis [13]. The activity of zooplankton largely will, in most cases, determine the particulate organic carbon deposited in seawater [14], and about 0.5 Pg (C)/A of the total CO_2_ they capture is deposited and stored on the seafloor [15]. Globally, marine algae are a kind of marine organism with a strong carbon sink function. Phototrophic dinoflagellates, cryptomonads, diatoms, heterotrophic bacteria, and heterotrophic dinoflagellates algae can efficiently absorb and store carbon from the atmosphere [16]. Alpert et al. (1992) predicted the algal marine carbon sink capacity in the continental shelf, and the research results showed that it was 0.7 GtC/a at a breeding cost of 300 USD/Tc/a [17]. Shellfish also play an important role in the marine carbon sink system. They absorb bicarbonate (HCO_3_^−^) in seawater to form calcium carbonate (CaCO_3_) to achieve carbon sequestration. The reaction equation is: Ca^2+^ + 2HCO_3_^−^ = CaCO_3_ + CO_2_ + H_2_O [18]. Therefore, the combined cultivation of shellfish and marine algae, as the most important patterns of the marine carbon sink fisheries, will effectively improve the economic and environmental benefits of marine fishery in this region [19].

On this basis, Lai et al. (2022) took nine coastal provinces in China as the research samples and used the material quality assessment method to estimate the biological carbon sink capacity of marine algae and shellfish [20]. Wu and Li (2022) quantitatively measured the current marine carbon sink capacity generated by the marine industry in ten coastal provinces (autonomous regions) in China from 2008 to 2019 and analyzed its spatio-temporal evolution characteristics from the national and provincial perspectives [21].

### 2.2. Research on Digital Inclusive Finance

Some of the studies on digital inclusive finance focused on the evaluation and measurement of its development level. Beck et al. (2007) estimated the level of financial services in 99 countries worldwide by constructing an evaluation system that included eight indicators such as the number of automated teller machines (ATMs) per 1000 km^2^, the number of ATMs per 100,000 people, and the number of bank branches per 1000 km^2^ and 100,000 people [22]. Sarma and Pais (2011) first measured the financial inclusion development index in 45 countries based on three dimensions: geographical penetration, service availability, and product usage [23]. Demirguec-Kunt and Klapper (2013) built an index system based on the customer terminal for the first time and assessed the level of financial inclusion in 148 countries using data on household savings, lending, payment, and risk control [24]. At present, a more representative indicator for measuring the level of development of digital inclusive finance in China is the Peking University Digital Financial Inclusion Index (2011–2020) released by the Institute of Digital Finance Peking University and Ant group in 2021, which presents the development trend of digital financial inclusion in different regions of China [25].

On this basis, numerous studies in the literature have examined the impacts of digital financial inclusion on the agricultural economy in the context of the new era. Combining digital technology and financial services to achieve financial innovation plays a crucial role in the transformation and upgrading of agribusiness [26]. Gao et al. (2022) found that the development of digital inclusive finance was an important path to motivate agricultural technology innovation and industrial structure optimization in China, which helped drive Chinese agriculture to improve agricultural total factor productivity and achieve high-quality development [27]. Based on a panel dataset of Chinese provinces from 2011 to 2019, Guo et al. (2022) constructed agricultural green development indicators from four dimensions of resource savings, environmental protection, ecological conservation, and quality industrialization and confirmed that digital inclusive finance have effectively promoted the green development of Chinese agriculture [28]. Zhong et al. (2022) found that the digital financial inclusion in China had a promoting effect on China’s agricultural technological progress, which in turn inhibited the intensity of agricultural carbon emissions in China [29]. Fang and Zhang (2021) demonstrated that during the COVID-19 epidemic, digital financial inclusion effectively served to keep the local agricultural supply chain stable through three mechanisms: financial broadening, financial deepening, and digitization of financial services [30]. Zhao et al. (2022) announced that the emergence of digital finance provided small farmers in China with a new way to obtain credit and alleviate their credit constraints, which may exert an impact on the adoption of sustainable agricultural practices [31]. Adegbite and Machethe (2020), using time-series data from 2011–2017 in Nigeria, showed that digital financial inclusion or other forms of financial innovation would greatly accelerate the process of achieving sustainable development in the smallholder agriculture of Nigeria [32].

In addition, regarding the DIFSE, Zhou and Zheng (2021) used the DEA method to assess the efficiency of fintech serving China’s real economy and found that the real economic growth, the level of urbanization, and the degree of openness were the significant factors affecting this efficiency [33]. Chen and Wang (2022) evaluated the financing efficiency of China’s strategic emerging industries using the DEA method, and confirmed that the asset–liability ratio, financial expenses, cash ratio and financing efficiency were negatively correlated, while the net asset income was a positively affecting factor [34]. Based on Banker, Charnes and Cooper’s improved data envelopment analysis model (BCC-DEA), which decomposed technical efficiency into pure technical efficiency and scale efficiency to calculate the efficiency value under variable payoffs of scale, Jin et al. (2021) measured the poverty reduction efficiency of digital inclusive finance in China by taking the coverage breadth and usage depth of digital inclusive finance as the inputs and the gap between the rich and poor as the outputs [35,36]. On this basis, Ren and Wang (2021) measured the support efficiency and Malmquist index of digital inclusive finance for poverty alleviation in western China from 2011–2018 using the DEA-Tobit model, and the development level of the primary industry, the degree of government support, the development level of the Internet, the education level of the population and the income gap between urban and rural areas were proven to be the significant determinants affecting the efficiency [37].

To sum up, the current research on marine carbon sink fisheries is still in its initial stage, and from the research point of view, most of them are focused on theory, mechanism, and carbon sink measurement, with many issues needing to be further explored. In the field of research on digital inclusive finance, the literature on issues such as level measurement, agricultural transformation and upgrading effect, and measurement of the DIFSE has been enriched. However, few scholars have examined and analyzed the role of digital inclusive finance in marine carbon sink fisheries, which is also part of the agricultural sector. In addition, DEA models have the drawback of not being compatible with radial and non-radial data. As a result, financial support efficiency studies using these models often do not yield accurate results. In addition, scholars tend to select the determinants of financial support efficiency subjectively, which inevitably makes the construction of the determinant system less comprehensive.

Therefore, using Deng’s grey relational analysis model, based on data in nine coastal provinces of China from 2011 to 2019, we empirically examine the relational coefficients between the digital inclusive finance level and China’s marine carbon sinks. Then, the Super-EBM is applied to evaluate the DIFSE for marine carbon sink fisheries. On this basis, the determinants affecting DIFSE, which are selected via grounded theory, are discussed with the help of the Tobit model.

## 3. An Analytical Framework Based on Grounded Theory

The DIFSE for marine carbon sink fisheries is influenced by a wide range of factors, while there are few existing research results on this issue. In addition to traditional production factor inputs that can impact the efficiency of capital use in carbon sink fisheries, there are also unidentified and unproven factors that will shape the DIFSE for marine carbon sink fisheries. Therefore, on the basis of previous studies, this paper tries to establish an analytical framework for the determinants of DIFSE with the help of grounded theory [38]. Grounded theory is a qualitative research method, which sets out to form the concepts, categories, and connections from data and ultimately establishes an analytical framework from the bottom up [39].

### 3.1. Information Sources

The information in this section is collected through Internet channels, including media reports, official data, journal articles, government documents, etc. Firstly, this paper identifies the search engines such as Bing, Baidu, Sogou, and Google as well as databases such as Web of Science, and China National Knowledge Infrastructure for data collection. Secondly, we take the terms “marine carbon sink fishery” and “digital inclusive finance” as keywords. Through the search engine, a total of more than 900 pieces of information were collected, while in the literature database, 78 pieces of information were searched. The information retrieved through the two methods mentioned above is highly “intrusive”; that is, there are many materials whose content is not relevant to marine carbon sink fisheries and digital inclusive finance. Therefore, based on the manual filtration, this paper finally obtains more than 500 information materials after screening. Data collection is an ongoing project and cannot be accomplished in one stroke. When the meaning of concepts and categories is unclear, or when the logic of theoretical relationships is not rigorous, it is necessary to collect the latest data in a timely manner.

### 3.2. Three-Level Coding

Level 1 coding, also known as open coding, is the process of conceptualizing and then categorizing all the information collected, in which the researcher needs to be open-minded, needs to try to suspend personal biases and stereotypes of the research community, and needs to code all the information as it is initially presented. It is an operational procedure of breaking up all the information, giving the concepts separately, and then putting them back together in a new way. The purpose of coding is to discover concepts and categories from the collected data, to identify the attributes and dimensions of the categories, and to finally name and categorize the subjects studied. According to the requirements of level 1 coding, we coded more than 500 pieces of information obtained from the screening, and finally abstracted several concepts and categories.

When analyzing concepts or categories in relation to each other, the researcher must not only consider the literal connections between them, but also explore the intentions and motivations of the research that expressed them, and the context and the socio-cultural background in which they lived also need to be taken into account.

Level 2 coding, also known as axial coding, is a process in which the main task is to identify and establish the links between concepts or categories in order to reflect the organic connection between the various parts of the collected data. These connections can be cause–effect oriented, temporal sequential, semantic, similarity, differential, reciprocity, structural, etc. In the level 2 coding process, the researcher should analyze only one concept in depth and look for various correlations around this concept, called “axis”. This paper takes the DIFSE for marine carbon sink fisheries as the core concept, around which the following three main categories are distilled.

Firstly, for resource input, in order to ensure the efficiency in the use of funds for marine carbon sink fisheries, it is necessary to invest in basic input productive factors such as breeding sites, labor, seedling, fishing vessel, and other fishery infrastructure. In addition, fishery activities require breeders to master the necessary production skills and technology. Furthermore, the improved breeds, aquaculture equipment, and technology evolution all require investment in human and material resources for theoretical innovation and achievement transformation. Therefore, the promotion of fishery skills and the innovation of fishery technology are also important to achieve the healthy development of marine carbon sink fisheries. As a result, this paper takes productive factors, fixed assets (fishing vessels and other fishery infrastructure), promotion of fishery skill, and fishery technology as the content of the main category of resource input.

Secondly, economic fundamentals, the upgrading and optimization of the structure of the marine carbon sink fisheries, and the expansion of the economic scale reflect the healthy growth of the fishery industry. The improvement of the income level of the residents and the continuous popularization of the Internet will continuously inject momentum for the sustainable development of marine carbon sink fisheries. Therefore, this paper takes the four major concepts of output structure, economic scale, income level, and Internet coverage as the content of the main category of the economic fundamental.

Finally, there is the environmental impact. The pollution of the sea area has a significant effect on the farming process of marine carbon sink organisms, and in serious cases, it may even lead to fishery disasters and hinder the further development of carbon sink fisheries. At the same time, environmental regulation plays a policy-oriented role in marine carbon sink fisheries as part of the low-carbon economy. In addition, the appropriate use of aquaculture drugs to assist in the farming process can improve production efficiency, which in turn can enhance the efficiency of capital use. Therefore, this paper takes marine pollution, aquaculture drug, environmental regulation, and fishery disaster as the content of the main environmental impact category, and the specific refining process is shown in Figure 1.

In level 3 coding, also known as selective coding, the researcher should analyze and select the “core concept” among all the concepts. The “core concept” should be overarching compared to other conceptual categories and play a role in a manner the full-text outline for other categories. After a systematic analysis, this paper takes into account the three major categories of resource input, economic fundamental, and environmental impact, and distills the core concept of “determinants influencing the DIFSE for marine carbon sink fishery”. The level 3 coding process is shown in Figure 2.

Finally, we establish the analytical framework of determinants including three dimensions of resource input, economic fundamental, and environmental impact with the help of grounded theory. On this basis, we can empirically discuss below which factors will influence the DIFSE for marine carbon sink fisheries.

## 4. Materials and Methods

### 4.1. Empirical Models

#### 4.1.1. The Grey Relational Analysis Model

The data on the marine aquaculture (mariculture) industry in China are poorly available and do not meet the typical distribution characteristics of research samples, while the grey system theory, founded by the famous Chinese scholar, Professor Deng Julong, is a control theory for studying information-poor, small-sample, uncertain systems [40]. Therefore, we used the grey relational model to analyze the correlation between the development of digital inclusive finance and China’s marine carbon sink capacity. This can avoid the bias of the results caused by the sample not meeting the requirements of traditional ordinary least square (OLS) regression: large data volume and conforming to typical distribution characteristics.

Commonly used grey relational analysis models include Deng’s grey relational analysis model, the grey absolute relational analysis model, the grey relative relational analysis model, and the grey comprehensive relational analysis model. Deng’s grey relational analysis model is the earliest, most frequently used and classic grey correlation model [41]. Deng’s grey correlation is defined by the “distance” between two series: the smaller the distance, the greater the degree of correlation between them. The grey absolute relational analysis determines the similarity of two sequences by exploring the relationship between their absolute quantities. The grey relative relational analysis model evaluates the degree of correlation between two time series curves according to the proximity of their change trends: if the increments of the two series are equal or close in a certain period, the degree of correlation between the two series is relatively high. The grey comprehensive correlation includes both the grey absolute correlation and the grey relative correlation, the synthetic coefficient θ indicates the degree of emphasis on the two. In this paper, we chose Deng’s grey relational analysis model to analyze the correlation between digital inclusive finance and marine carbon sinks in China.

The steps of calculating Deng’s grey correlation are:(1)The reference sequence and the comparison sequence need to be determined.

The reference sequence.
(1)$$X_{0}=\left({x_{0}}^{\left(1\right)},{x_{0}}^{\left(2\right)},\ldots ,{x_{0}}^{\left(n\right)}\right)$$

The comparison sequence.
(2)$$X_{i}=\left({x_{i}}^{\left(1\right)},{x_{i}}^{\left(2\right)},\ldots ,{x_{i}}^{\left(n\right)}\right)$$
where, *i* = 1, 2, 3,…, *m*.

(2)Based on Equation (3), the grey relational coefficients of the reference sequence *X*_0_ and the comparison sequence *X_i_* at moment *t*: $\xi \left({x_{0}}^{\left(t\right)},{x_{i}}^{\left(t\right)}\right)$ are calculated.

(3)$$\begin{array}{c} \xi \left({x_{0}}^{\left(t\right)},{x_{i}}^{\left(t\right)}\right)\\ =\frac{\underset{i}{min}\underset{t}{min}\left|{x_{0}}^{\left(t\right)}-{x_{i}}^{\left(t\right)}\right|+\lambda \underset{i}{max}\underset{t}{max}\left|{x_{0}}^{\left(t\right)}-{x_{i}}^{\left(t\right)}\right|}{\left|{x_{0}}^{\left(t\right)}-{x_{i}}^{\left(t\right)}\right|+\lambda \underset{i}{max}\underset{t}{max}\left|{x_{0}}^{\left(t\right)}-{x_{i}}^{\left(t\right)}\right|} \end{array}$$
where λ is the resolution coefficient, and its value range is (0,1); generally, λ is taken as 0.5.

(3)The grey correlation between the reference sequence *X*_0_ and the comparison sequence *X_i_* is as follows.


(4)
$$\gamma \left(x_{0},x_{i}\right)=\frac{1}{n}\overset{n}{\underset{t=1}{\sum }}\xi \left({x_{0}}^{\left(t\right)},{x_{i}}^{\left(t\right)}\right)$$


#### 4.1.2. The Super-EBM Model

In 1978, Charnes et al. (1978) proposed the DEA method to compare the relative efficiency of multiple decision making units (DMUs) in providing similar services [42]. This method does not require the setting of a production function and therefore has the advantage of being applicable to the evaluation of efficiency in multiple scenarios. Traditional DEA models (CCR, BCC, SBM) only measure the efficiency of DMUs in radial or non-radial mode and are not compatible with radial and non-radial data [43]. However, in the assessment of DIFSE for marine carbon sink fisheries, radial and non-radial characteristics coexist in the growing process of input and output variables. To improve this deficiency, Tone and Tsutsui (2010) compiled the two different characteristics: radial and non-radial, into a composite model called “Epsilon-Based Measure” (EBM) [44]. However, the initial EBM model cannot compare the efficiency values of multiple DMUs at the production frontier, and it is illogical to set input orientation or output orientation. Therefore, combining with the super-efficiency DEA model proposed by Andersen and Petersen (1993), this paper adopts a non-oriented, variable payoffs of scale Super-EBM model to study the DIFSE for marine carbon sink fisheries in China [45]. The specific equation is shown below.
(5)$$r^{*}=min\frac{\theta +\varepsilon_{x}\sum_{i=1}^{m}\frac{{w_{i}}^{-}{s_{i}}^{-}}{x_{ik}}}{\varphi -\varepsilon_{y}\sum_{r=1}^{q}\frac{{w_{r}}^{+}{s_{r}}^{+}}{yrk}}\;s.t.\overset{n}{\underset{j=1,j\neq k}{\sum }}x_{ij}\lambda j-{s_{i}}^{-}\leq \theta x_{ik},i=1,2,\ldots ,m\;\overset{n}{\underset{j=1,j\neq k}{\sum }}\lambda j=1,\lambda \geq 0,{s_{i}}^{-}\geq 0,{s_{r}}^{+}\geq 0$$
where *r*^*^ is the efficiency value of the Super-EBM model with *m* inputs: $x_{ik}$ (*i* = 1, 2,…, *m*) and *q* outputs: $y_{rk}$ (*r* = 1, 2,…, *q*) for each DMU. ${s_{i}}^{-}$ represents the non-zero slack variables of input factors (input redundant variables), and ${s_{r}}^{+}$ represents the non-zero slack variables of output factors (output deficiency variables), and the following relationships $\overset{\_\_\_}{x_{ik}}=x_{ik}-{s_{i}}^{-}$ and $\overset{\_\_\_}{y_{rk}}=y_{rk}+sr^{+}$ exist, i.e., theoretically, to achieve the ideal situation, inputs should be reduced and outputs should be increased. ${w_{i}}^{-}$ and ${w_{r}}^{+}$ represent the weights of the input and output factor quantities, respectively. λ is the combination coefficient of the decision-making unit. ε is the key parameter in the EBM model, which represents the combination ratio of the radial model and non-radial model, and takes values in the range from 0 to 1. If ε = 0, the model is equivalent to the radial model, conversely, if ε = 1, the EBM model is equivalent to the SBM model. θ and φ are the radial partial efficiency values of *x*, *y* respectively.

#### 4.1.3. The Tobit Model

The Tobit model, also known as the restricted dependent variable model, proposed by Tobin in 1958, applies to the situation where the dependent variable takes continuous values but is subject to certain restrictions [46]. Since the DIFSE for marine carbon sink fisheries measured by the Super-EBM model is truncated, using OLS regression analysis would cause a lack of unbiasedness and consistency. As a result, this paper uses a panel Tobit model for regression analysis with the following equation.
(6)$$\gamma \left(x_{0},x_{i}\right)=\frac{1}{n}\overset{n}{\underset{t=1}{\sum }}\xi \left({x_{0}}^{\left(t\right)},{x_{i}}^{\left(t\right)}\right)y_{it}^{*}=\alpha +\beta^{T}x_{it}+\varepsilon_{it}\;y_{it}=\{\begin{array}{c} y_{it}^{*},y_{it}^{*}>0\\ 0,y_{it}^{*}\leq 0 \end{array}$$
where $y_{it}^{*}$ is unobservable, α is the vector of intercept terms, $\beta^{T}$ is the vector of parameters to be estimated, $x_{it}$ is the vector of independent variables, $y_{it}$ is the vector of dependent variable, and the random disturbance $\varepsilon_{it}$ is independent and follows a normal distribution.

### 4.2. Variables Selection

#### 4.2.1. Digital Inclusive Finance and Marine Carbon Sinks

(1)The index of digital inclusive finance

To measure the development level of digital inclusive finance, this paper applies the Digital Financial Inclusion Index mentioned above. The index, compiled by the Ant Financial Services Group and the Institute of Digital Finance Peking University since 2011 using a dimensionless approach, contains three dimensions: usage depth, coverage breadth, and degree of digitization, and it covers 31 provinces (autonomous regions) and 337 prefecture-level cities in China. Here, for an in-depth study of the impact of digital inclusive finance on marine carbon sinks, four sub-indicators of digital inclusive finance: usage depth, coverage breadth, degree of digitization, and credit operations, are also selected for analysis in this paper.

(2)Marine carbon sinks

Regarding China’s marine carbon sinks, this paper targets shellfish, and algae products from mariculture. The classification of carbon sinks achieved by algae mainly includes carbon sinks removed at harvest, dissolvable organic carbon (DOC), and particulate organic carbon (POC) transported to the seabed. The carbon sinks of shellfish consist mainly of carbon fixed in their shells and soft tissues that are removed at harvest. On this basis, drawing on previous studies, this paper calculates the marine carbon sink capacity of nine coastal provinces (autonomous regions) of China from 2011 to 2019 [47,48]. The variable design and descriptive statistics are shown in Table 1.

#### 4.2.2. The Output and Input Variables in the Measurement of the DIFSE for Marine Carbon Sink Fisheries

In terms of output variables, the indicator of China’s marine carbon sinks as described above is selected. In terms of input variables, referring to the study of Ren and Wang (2021), indicators are constructed from two dimensions of penetration and accessibility [37], and according to the Digital Financial Inclusion Index, the usage depth and the coverage breadth are selected based on penetration; based on accessibility, the variable of digital inclusive finance credit operations in China is selected. Among them, the breadth of digital inclusive financial coverage in China includes data such as the number of Alipay accounts per 10,000 people, the proportion of Alipay bound bank card users and the average number of bank cards bound to each Alipay account. China’s digital financial usage depth includes Alipay payment, transfer, wealth management, credit qualification and other business data. The credit operations mainly consist of data on both personal consumer loans and loans to small and micro firms made possible by Alipay. The specific characteristics of the variables are shown in Table 1.

In addition, to explore the regional heterogeneity in the grey relational coefficients and the DIFSE for marine carbon sink fisheries, this paper divides China’s coastal regions into northern, eastern, and southern marine economic circles based on the differences in the marine industrial agglomeration and the descriptions in the National Marine Economy Development Plan. The northern marine economic circle includes three provinces of Hebei, Liaoning, and Shandong; the eastern marine economic circle includes Zhejiang and Jiangsu; and the southern marine economic circle contains Fujian, Guangdong, Guangxi, and Hainan four provinces (autonomous regions).

#### 4.2.3. Determinants Affecting the DIFSE for Marine Carbon Sink Fisheries

Based on the analytical framework of determinants affecting the DIFSE for marine carbon sink fisheries established with the help of grounded theory, the following variables are selected from three dimensions of resource input, economic fundamental, and environmental impacts.

(1)Resource input

In this dimension, productive factors (*factors*), promotion of fishery skill (*skill*), and fishery technology (*tech*) are selected for empirical investigation. The productive factors include the breeding sites for marine carbon sink fisheries, the quantity of marine carbon sink fishery labor, the number of seedlings, the fixed assets, and the intermediate consumption. The variable of breeding sites is characterized by the area of shellfish and algae mariculture sites. The quantity of labor force is measured by the number of professional practitioners of the carbon sink fishery. The variable of seedlings is characterized by the number of shellfish and algae seedlings. The power of fishing vessels for marine carbon sink fisheries is selected to measure the variable of fixed assets. With the idea provided by Qin et al. (2018), the intermediate consumption of marine carbon sink fisheries can be converted from the intermediate consumption value of the fishery and can be obtained by excluding the price factor according to the price index of agricultural means of production [49]. On this basis, this paper applies the Min–Max standardization method, i.e., $x^{*}=\left(x-min\right)/\left(max-min\right)$, to standardize each indicator of the above production factors, and then sums up the values of the standardized indicators to obtain the data of productive factors (*factors*).

Similarly, the promotion of fishery skill (*skill*) includes the training and promotion of skill. Therefore, two sets of data, aquatic skill business promotion funds and the number of fisherman skillful training, are standardized using the Min–Max standardization method, and the standardized values are summed to characterize the level of fishery skill promotion. Drawing on the research of Lin et al. (2019), the sum of the standardized values of the number of research papers related to marine carbon sink fisheries and the number of related patents is selected to characterize the variable of fishery technology (*tech*) in this paper [50].

(2)Economic fundamental

In this dimension, the output structure (*structure*), regional income level (*income*), and Internet coverage (*IC*) are selected for examination in this paper. Here, the output structure refers to the proportion of marine carbon sink fishery production to total mariculture production, and the larger the ratio, the more the mariculture structure can meet the requirements of a low-carbon economy. Regarding the income level, disposable income is closest to the concept of income as generally understood in economics [51]. The income levels are generally reflected by consumption, and the most important determinant of consumption is per capita disposable income. Therefore, referring to the study of Hu et al. (2022), this paper uses the value of regional per capita disposable income to measure this variable [52]. Based on the study of Guo and Luo (2016), the variable of Internet coverage (*IC*) is characterized by the ratio of regional Internet broadband access users to the number of the regional resident population [53].

(3)Environmental impacts

The environmental factors that may affect the efficiency of fund utilization for marine carbon sink fisheries may include marine pollution (*pollution*), environmental regulation (*ER*), fishery disaster (*disaster*), and aquaculture drug (*drug*). In this paper, we chose marine wastewater discharge to characterize the degree of marine pollution. The number of marine pollution control investment projects is used to measure the intensity of environmental regulation, based on the study of Qian and Liu (2013) [54]. Fisheries disasters usually include diseases, meteorological disasters, etc. However, meteorological disasters cannot be prevented and controlled artificially; thus, this paper measures fishery disasters as the proportion of the affected area of breeding sites due to diseases to the total area. Finally, on the use of aquaculture drug, this variable is expressed in this paper as the share of the output value of fishery drugs in the total output value of mariculture.

Since it is difficult to obtain the data on marine carbon sink fisheries directly, this paper draws on the output value ratio conversion method and area ratio conversion method, and sets the following conversion coefficients: α = area of marine carbon sink fishery breeding sites/total area of mariculture breeding sites, β = output value of marine carbon sink fishery/total output value of the fishery [55,56]. The labor input, power of fishing vessels, intermediate consumption, skill promotion, technology, drug use, and affected area of marine carbon sink fisheries were all obtained by multiplying the corresponding raw data with the conversion coefficients described previously. The variable design and descriptive statistics are shown in Table 2 below.

### 4.3. Data

For all the variables mentioned above, this paper takes the panel data of nine coastal provinces (autonomous regions) of China from 2011 to 2019 as the research sample. In terms of the digital inclusive finance and its sub-indicators, the data source is the literature of “Measuring China’s Digital Financial Inclusion: Index Compilation and Spatial Characteristics” [25], and data used to calculate China’s marine carbon sinks are from the China Fishery Statistical Yearbook.

In addition, the original data of fishery professional practitioners, breeding sites area, number of seedlings, power of fishing vessels, aquatic skill business promotion funds, number of fisherman skillful training, number of fishery-related research papers and related patents, marine carbon sink fishery production and output value, mariculture production and output value, affected breeding sites area, and fishery drug output value are all from the China Fishery Statistical Yearbook. The data on intermediate consumption of fishery are sourced from China Rural Statistical Yearbook. The price index of agricultural means of production and regional per capita disposable income are sourced from China Statistical Yearbook. The sources of the two sets of data on regional Internet broadband access users and the regional resident population are the China Internet Network Information Center and the local statistics department, respectively. The source of marine wastewater discharge data is the China Marine Ecological Environment Status Bulletin, and the number of marine pollution control investment projects is from the China Marine Statistical Yearbook over the years.

### 4.4. Statistical Analysis

The results of the grey relational analysis model were obtained on Grey Modelling 6.0 Software (Grey System Research Institute, Nanjing University of Aeronautics and Astronautics, Nanjing, Chian). The Super-EBM model was conducted on MaxDEA 7.0 software (Beijing Ruiwo Maidi Software Co., Ltd., Beijing, China), and the Tobit regression results were obtained by Stata 17.0 software (StataCorp LLC, College Station, TX, USA).

## 5. Results

### 5.1. Whether There Is a Correlation between the Level of Digital Inclusive Finance and China’s Marine Carbon Sinks

The initial series of digital inclusive finance, its sub-indicators, and China’s marine carbon sinks were imported in Grey Modelling Software to obtain Deng’s grey correlation between each factor of digital inclusive finance and the parent factor marine carbon sinks in the three marine economic circles. The results are shown in Table 3.

From Table 3, it can be seen that the correlation between each of the digital inclusive finance indicators and marine carbon sinks is above 0.5, exhibiting a positive correlation between the level of digital inclusive finance and the development of China’s marine carbon sink fisheries, which confirms that digital inclusive finance may have alleviated the financing constraints of marine carbon sink fisheries from the other aspect. Further, the development of digital inclusive finance has helped the green transformation of the fishery industry and has contributed significantly to the accelerated completion of a low-carbon economy in China, which is similar to the conclusion reached by Guo et al. (2022) using a spatial Durbin model [28]. In addition, there is a large degree of variation in the relational coefficients between different sub-indicators of digital inclusive finance and marine carbon sinks.

From the results in the northern marine economic circle, the correlation between the indicators of digital inclusive finance and marine carbon sinks ranged from 0.5010 to 0.8990. The mean correlation between the total indicators of digital inclusive finance *DIF* and the parent series in the northern marine economic circle is 0.6222, and the two sub-indicators of credit operations and usage depth have the highest correlation with marine carbon sinks, with mean values of 0.8335 and 0.6785, respectively. The relational coefficient mean values of 0.5994 and 0.5700 for coverage breadth and degree of digitalization, respectively, are relatively low, reflecting that, compared with mere expansion of the breadth of digital inclusive financial coverage, the depth of digital inclusive financial services and the continuous improvement of credit business provide better support effects for marine carbon sink fisheries in the three regions of Hebei, Liaoning, and Shandong.

From the results in the eastern marine economic circle, the overall correlation between each indicator and the parent series is high, taking values between 0.5074 and 0.9318. The mean value of correlation between the total indicators of digital inclusive finance *DIF* and marine carbon sinks generated in Zhejiang and Jiangsu reaches 0.8375, showing a significant strong positive correlation. Among them, the strongest correlation is for credit operations, with an average grey relational coefficient of 0.9301, followed by the breadth of digital inclusive financial coverage and usage depth, both of which have similar correlations with the parent series, with mean values of 0.8713 and 0.8647, respectively. The above results reflect that in Jiangsu and Zhejiang provinces, the development of digital inclusive finance credit services can directly alleviate the problem of expensive and difficult financing for marine carbon sink fisheries, and thus promote the healthy development of this industry. In addition, the digital technology development in the eastern region is more mature, the infrastructure construction is more perfect, and the residents have a higher degree of awareness and trust in digital financial inclusion, resulting in the expansion of the breadth of digital inclusive financial coverage that often increases user stickiness and leads users to use credit services, thus effectively enhancing its financial support effect. Furthermore, the correlation for the degree of digitalization is relatively low, with a mean value of 0.5078. The reason for this is that digitalization may mainly reflect the impact of digital technology on the mobility, affordability, and convenience of residents’ lives. The radiation effect on the development of marine carbon sink fisheries is not obvious.

From the results in the southern marine economic circle, the correlation of each indicator ranges from 0.5039 to 0.8425. The mean value of the correlation between the total digital inclusive finance indicator *DIF* and marine carbon sink in the four southern provinces is 0.6238, similar to the results in the northern marine circle. Credit operations and usage depth have the highest relational coefficients, with mean values of 0.8179 and 0.6863, respectively. The relational coefficients of the degree of digitalization and the coverage breadth are 0.5938 and 0.5877, respectively, which are relatively low. It may be due to the fact that in the southern region, because of the lack of awareness, end users may prefer to view digital inclusive finance as a convenient mobile payment tool rather than attaching importance to its credit business services, resulting in the neglect of its inclusive function, which in turn leads to a small increase in the support effect when simply enhancing the breadth of coverage of digital inclusive finance.

To sum up, firstly, in general, digital inclusive finance plays a strong role in promoting the marine carbon sink fisheries, and the development of digital inclusive finance credit operations can significantly support the healthy development of the marine carbon sink fisheries in coastal provinces across the country, while the degree of digitalization is more a reflection of the convenience and mobility of people’s lives due to the technological changes, resulting in a generally low correlation between this indicator and marine carbon sink. Secondly, through the inter-regional comparison, the correlation between digital inclusive finance and marine carbon sinks is highest in the eastern marine economic circle, followed by the northern and southern marine economic circles. This result may stem from the fact that digital financial inclusion is the most mature and longest developed in the eastern region compared to the north and south, with the most active credit operations applied to micro- and small enterprises. Finally, in the southern and northern circles, the usage depth is highly correlated with the parent series, while the coverage breadth is relatively less correlated with marine carbon sinks, and the difference between them is obvious. However, in the eastern circle, the two correlations have similar values, and the difference is small, with the mean difference controlled at 0.0066. This result reflects that in the southern and northern marine economic circles, the potential of digital inclusive finance to support marine carbon sink fisheries still needs to be further explored. The user guidance mechanism should be further improved to enhance the audience’s cognition and understanding of digital inclusive finance, so that the expansion of coverage can be effectively transformed into the increased use of credit services. On this basis, we can rely on digital inclusive finance credit business to break the status quo that “marine enterprises” are difficult to be financed due to traditional financial discrimination, improve the efficiency of matching supply and demand funds, and ensure the healthy development of marine carbon sink fisheries.

### 5.2. The DIFSE for Marine Carbon Sink Fisheries

The breadth coverage, usage depth, and credit operations of digital inclusive finance are selected as input variables, and the indicator of marine carbon sinks is selected as the output variable. On this basis, using MaxDEA software, according to Equation (5), we measured the DIFSE for the marine carbon sink fishery *Score* in China’s nine coastal provinces (autonomous regions) from 2011 to 2019, and the results are shown in Table 4.

(1)Analysis of the time variation in the DIFSE for marine carbon sink fisheries

As can be seen in Table 4, from 2011 to 2019, the DIFSE for China’s marine carbon sink fisheries ranged from 0.4390 to 1.3888, and the DIFSE in the three provinces located in the northern marine economic circle: Hebei, Liaoning, and Shandong, decreased by about 8% over 9 years, while the efficiency value in eastern coastal provinces and the southern coastal provinces both showed an upward trend, with an increase of about 70% and 25%, respectively. Among them, Shandong province, which has the highest average DIFSE value over the 9 years, has the largest decline in DIFSE, with approximately 20% less in 2019 compared to 2011, but its value is still greater than 1 in 2019, indicating that the role of digital inclusive finance in supporting marine carbon sink fisheries is still efficient in Shandong province, while the province of Zhejiang, which has the lowest mean DIFSE value, increased its efficiency by 77% to 0.7782 in 2019 compared to 2011. It can be seen that the efficiency growth rate in low-efficiency provinces is significantly higher than that in high-efficiency provinces. The effectiveness of China’s digital inclusive finance support for marine carbon sink fisheries has expanded year by year. The mean value of DIFSE for marine carbon sink fisheries in nine provinces increased from 0.8257 in 2011 to 0.9489 in 2019, with an increase of 14%. More provinces were able to move closer to their production frontiers compared to previous years.

(2)Analysis of the regional differences in the DIFSE for marine carbon sink fisheries

It can also be seen from Table 4 that the top three rankings of the mean value of the DIFSE for marine carbon sink fisheries are Shandong (1.1770), Hebei (1.0139), and Guangxi (1.0125). Among them, Shandong and Hebei are both parts of the northern marine economic circle, while Guangxi is in the southern marine economic circle. The last three places in the ranking of efficiency average are Guangdong (0.7773), Jiangsu (0.7490), and Zhejiang (0.6769). Among them, Guangdong is in the southern marine economic circle, while Jiangsu and Zhejiang are both located in the eastern marine economic circle. The efficiency in each province is divided according to the three marine economic circles, and the average efficiency of each circle is calculated to obtain Figure 3. It was found that in the northern marine economic circle, digital inclusive financial support is most efficient, with efficiency values between 1.0283 and 1.1472, followed by the southern marine economic circle, with efficiency values between 0.7604 and 0.9501, and the eastern circle has the lowest efficiency, with its value ranging from 0.4740 to 0.8229. To sum up, the efficiency of China’s digital inclusive financial support shows a significant regional distribution imbalance, with the highest in the Huanghai and Bohai Seas, the middle in the South China Sea, and the lowest in the East China Sea. An obvious V-shaped spatial distribution feature exists from north to south.

First, from an overall perspective, the average level of China’s DIFSE for marine carbon sink fisheries is high, but there are still scopes for improvement. The mean values of DIFSE in most provinces are around 0.9, indicating that the development of marine carbon sink fisheries and digital financial inclusion has not yet achieved an ideal synergistic coupling, and the pressure to transform the production mode and adjust the structure of China’s marine carbon sink fisheries will exist for a long time.

Second, from the perspective of the marine economic circle, the mean values of DIFSE for marine carbon sink fisheries in Shandong and Hebei, located in the Huanghai and Bohai Sea region, reach 1.1770 and 1.0140, respectively, and the DIFSE value of Guangxi in the South China Sea region is 1.0125. In these above three provinces, digital inclusive finance is more efficient in their support. Shandong, Hebei, and Guangxi have gradually entered into a new track of digital inclusive financial support for the sustainable development of carbon sink fisheries. Specifically, in 2018, the Guangxi Provincial People’s Government released the “Guangxi Three-Year Operation Program for Water Pollution Prevention and Control”, which restricts baiting net cage culture in offshore areas and actively develops shallow sea shellfish and algae aquaculture, reflecting the determination to win the battle against sea pollution and achieve a green low-carbon economy. Thus, a higher DIFSE is achieved through the output path. Shandong, as a demonstration area of marine pasture and the birthplace of shellfish and algae aquaculture, has the characteristics of a large production of shellfish and algae, and a high degree of intensification, and at the end of 2013, Hebei unveiled “Guidelines on Promoting the Sustainable Development of Marine Fishery” to accelerate the adjustment of mariculture patterns and promote standardized ecological and healthy breeding modes such as factory breeding, marine pasture, and ecological stereoscopic polyculture. In 2014, a total of four marine pastures were built with an area of 173 hectares, and 6.1 million shellfish and algae seedlings were transplanted and sown. Therefore, Hebei and Shandong are able to ensure that the input of digital inclusive finance brings a sufficient amount of carbon sink output. Zhejiang and Jiangsu in the East China Sea region have the lowest mean DIFSE values of 0.6769 and 0.7490.

The reason for the difference from the previous grey correlation results is that although digital inclusive finance is highly developed in the eastern region with higher relational coefficients, this more reflects the willingness of fishermen to use digital inclusive finance to help finance marine carbon sink fisheries. The leading industries of Zhejiang and Jiangsu fisheries are both economic fish cultures, with small output of shellfish and algae mariculture, which indicates that the marine carbon sink fisheries have failed to form the economies of scale. The marine carbon sink fisheries in the above two provinces have not attracted sufficient attention, leading to the backwardness of shellfish and algae mariculture patterns, reducing their capital utilization efficiency, and thus resulting in a low range of DIFSE, which is also confirmed by the research of Zhang et al. (2020) [57]. On the contrary, in Shandong, Hebei and Guangxi, the correlation between digital inclusive finance and marine carbon sinks is relatively low, reflecting that fisherman may make less use of the financing function of digital inclusive finance. However, with the advantage of their advanced patterns and the formation of economies of scale, they are more efficient in utilizing the financing support received through digital inclusive finance, thus placing the DIFSE in a higher range.

In addition, in order to more intuitively highlight the spatial and temporal characteristics of the DIFSE for China’s marine carbon sink industry, drawing on the existing research, this paper divides the efficiency values into five levels: lower efficiency (0–0.5000), low efficiency (0.5001–0.6000), middle efficiency (0.6001–0.7000), high efficiency (0.7001–0.9000), and higher efficiency (0.9001 and above) [58]. On this basis, we are able to map the spatial and temporal distribution of inter-provincial DIFSE using ArcGIS 10.8 software, as follows.

It can be seen from Figure 4 that the DIFSE for marine carbon sink fisheries was at lower, low, high, and higher levels in most provinces in 2011. Higher efficiency level areas included four provinces, Hebei, Liaoning, Shandong, and Guangxi. Fujian and Hainan were in high-efficiency level areas. Jiangsu and Guangdong had low-efficiency levels, while the DIFSE in Zhejiang province was in the lower range. From north to south, along with the color banding from dark to light to dark, the DIFSE for marine carbon sink fisheries went through a process of “first decreasing and then increasing”, which intuitively reflected the V-shaped characteristics of the distribution.

In 2015, efficiency levels had improved overall in China’s coastal provinces compared to 2011. Hebei, Liaoning, Shandong, and Guangxi remained at the higher-efficiency level, while Hainan Province rose one level to be among the most efficient provinces. Three regions were at the high level, namely Jiangsu, Guangdong, and Fujian, with Jiangsu, and Guangdong both moving up two levels, and Fujian already at the high-efficiency level in 2011. In addition, Zhejiang also rose two efficiency levels, from low to middle. Based on the perspective of the marine economic circle, the northern circle was in the leading position, the south was in the second place, and the eastern marine economic circle was still lagging behind the first two in terms of overall efficiency value, but its efficiency was growing faster and improving significantly. As of 2015, in all provinces, the DIFSE had reached above-middle levels, and the spatial distribution pattern of high-efficiency clustering had begun to take shape.

In 2019, the efficiency level in each province was further improved, with all nine coastal provinces (autonomous regions) achieving the high-efficiency level, and the coverage of high-efficiency agglomeration zones was expanded to further play a synergistic role. Zhejiang and Fujian both rose one level, to high-efficiency and higher-efficiency levels respectively. Two provinces that were at the high-efficiency level for the four years 2015–2019: Jiangsu and Guangdong, also saw an increase in their efficiency values. The efficiency value in Guangdong achieved an increase of 4.9633% compared to 2015, and DIFSE in Jiangsu rose by about 6.9086% from 0.7754 to 0.8290. The inter-circle efficiency differences still existed, but the gap was narrowed and the divergence of efficiency distribution was greatly alleviated, reflecting the further improvement and expansion of the effect of digital inclusive finance on supporting carbon sink fisheries.

### 5.3. What Factors Will Affect the DIFSE for Marine Carbon Sink Fisheries

In this paper, via the grounded theory, ten determinants were selected from three dimensions: resource input, economic fundamental, and environmental impact, and their regression coefficients with the DIFSE for marine carbon sink fisheries measured above are estimated. Based on Equation (6), we performed the regression analysis using Stata 17.0, and the sample data were panel data for nine coastal provinces from 2011–2019. In addition, to linearize the trend of each series and to eliminate their heteroskedasticity, the natural logarithm was taken for the series *income*, *pollution*, and *ER*. The regression results are shown in Table 5.

(1)In terms of the resource input

Firstly, the empirical results demonstrate that every 1% increase in productive factor inputs can increase the DIFSE for marine carbon sink fisheries by 0.0971%. This indicates that sufficient productive factor inputs can ensure that the credit financing inflow of digital inclusive finance helps marine carbon sink fisheries increase the amount of carbon sink output, which in turn improves the DIFSE. Secondly, both the promotion of fishery skill and fishery technology are significantly positive determinants of DIFSE. These results are consistent with the findings of Elhendy and Alkahtani (2012) [59]. For every 1% increase in the above two indicators, the DIFSE rises by 0.1803% and 0.0864%, respectively. The diffusion of advanced production skill can significantly reduce the input–output ratio of marine carbon sink fisheries, shorten the time required for the production process, and improve the speed and quality of output, thus enabling the same digital inclusive financial credit service provision to produce more marine carbon sinks, resulting in improved efficiency. The improvement of fishery technology has a positive effect on the DIFSE, reflecting that the theoretical innovation in the field of carbon sink fisheries can be practically transformed into productivity in reality, and the smooth transformation process of technical achievements can often realize the selection and breeding of improved species, optimize and upgrade breeding equipment, and improve production efficiency, thus making it possible to obtain a higher output of marine carbon sinks with less digital financial services and enhance the DIFSE.

(2)In terms of the economic fundamental

Firstly, the output structure of marine carbon sink fisheries has a significantly negative impact on the DIFSE for marine carbon sink fisheries in the local area, and for every 1% increase in the production share, the efficiency value will decrease by 0.2204%. Reflecting the relatively large scale of economic development of China’s marine carbon sink fisheries at present, it is in the stage of diminishing marginal productivity, and continuing to expand the output share will reduce the efficiency of resource utilization. The diseconomies of scale lead to the fact that the amount of inputs and outputs cannot increase in the same proportion, and the level of outputs will lag behind the level of factor inputs, thus weakening the support effect of digital inclusive financial credit services for carbon sink fisheries and reducing the efficiency value. Secondly, the income level is a significantly negative factor for the DIFSE, and for every 1% increase in the income level of residents, the DIFSE will decrease by 0.2288%. This confirms the results of Liang and Yang (2019), and Jin et al. (2017), namely that China is still in the first half of the Environmental Kuznets inverted U-curve (EKC) [60,61]. The increase in residents’ income level is accompanied by the deterioration of environmental quality, and thus the healthy development of marine carbon sink fisheries often goes unnoticed and faces numerous obstacles, which in turn reduces the effectiveness of digital inclusive finance in supporting this industry. Besides, in view of the large number of abbreviations in this paper, a table of abbreviation definitions is included to facilitate the reader’s understanding (see Appendix A). Finally, Internet coverage has a significantly positive effect on the DIFSE, and every 1% increase in this indicator will improve the efficiency value by 0.6593%. This result is consistent with the findings of Chen et al. (2021) that the popularization of the Internet and the continued improvement of digital technology infrastructure will further break down the information barriers, alleviate the problem of information mismatch, and enhance the efficiency of marine carbon sink fisheries financing and capital utilization [62].

(3)In terms of environmental impact

Firstly, marine pollution has a significant negative impact on the DIFSE for marine carbon sink fisheries. Each 1% increase in sea area wastewater discharge will decrease the DIFSE by 0.0886%. The results are similar to the conclusions obtained by Hai and Speelman (2020) [63]. The increase in marine pollution is often a reflection of a country’s tendency to prioritize heavy industry and the lack of environmental protection awareness among its people [64,65]. Both of these causes can reflect the failure of marine carbon sink fisheries to attract widespread attention as an important component of the low-carbon economy. The ecological carbon sequestration function of shellfish and algae is ignored, which hinders the development of carbon sink fisheries and reduces the support effect of digital inclusive finance. Secondly, fishery disasters exhibit a significant negative effect on the DIFSE, and every 1% increase in this indicator will decrease the efficiency value by 0.8219%, reflecting that fishery disasters, in whatever form, will impede the smooth running of the culture process, leading to loss of shellfish and algae products and lower output, which in turn detract from the DIFSE for marine carbon sink fisheries.

## 6. Conclusions and Discussion

### 6.1. Research Conclusions

This paper measures the correlation between the development of digital inclusive finance and the capacity of marine carbon sinks in the three marine economic circles of China through Deng’s grey relational analysis model. Subsequently, we further measure the DIFSE for marine carbon sink fisheries in nine coastal provinces of China using the Super-EBM model, and the determinants of DIFSE are investigated based on the Tobit regression model. The conclusions are as follows. First, digital inclusive finance exhibits a significant effect of support on the marine carbon sink fisheries. Second, the financial support effect of digital inclusive finance on marine carbon sink fisheries in China is strengthening year by year, the gap in efficiency values between regions is being further narrowed, and more decision-making units are able to move closer to their production frontiers than before. Among them, the northern marine economic circle is currently the most efficient, followed by the southern marine economic circle and the eastern marine economic circle. Third, productive factor input, promotion of fishery skill, fishery technology, and Internet coverage have positive promoting effects on DIFSE, while the output structure, income level, fishery disaster, and marine pollution, on the contrary, have significant inhibiting effects on the DIFSE.

### 6.2. Political Implications

This paper points out the direction for further development of China’s marine carbon sink fishery economy in the future through a series of empirical studies. Digital inclusive finance should be regarded as an important opportunity to break the dilemma of fishermen’s financing constraints. First, it is necessary to clarify the demand for financial support in the marine carbon sink industry and establish the main position of digital inclusive finance in credit services for small and micro-enterprises, so as to improve the financial support service system, broaden the financing channels for marine carbon sink fisheries, and ensure that the carbon sink fisheries, as an important part of the low-carbon economy, receive sufficient financial support to achieve stable and healthy development. In addition, the eastern marine economic circle has a sound and good digital infrastructure, but the DIFSE for marine carbon sink fisheries is relatively low. To address this phenomenon, the government should deeply recognize the potential of marine carbon sink fisheries in accelerating the achievement of sustainable development goals, and it should guide fishermen to gradually focus on shellfish and algae mariculture through subsidies, tax breaks, and other measures. In this way, the advantages of the more mature development of digital inclusive finance in the eastern region can be fully utilized to amplify the effect of its support for marine carbon sink fisheries. The digital inclusive finance support in the northern and southern marine economic circle is relatively efficient, but it is still necessary to further strengthen the application of digital inclusive finance in the field of small and micro-credit and guarantee sufficient credit support for marine carbon sink fisheries. Finally, in view of the results of the factors influencing the DIFSE, the Chinese authorities should designate policies according to local conditions and should accelerate the process of achieving the goals of “carbon peak” and “carbon neutrality” by continuously improving the digital technology infrastructure, actively combating marine pollution, enhancing people’s awareness of environmental protection and low carbon, building regional mariculture skill exchange platforms, and by increasing the investment in scientific research and technology invention in regions with weak levels of carbon sink fishery technology.

### 6.3. Limitations and Future Research Directions

At present, related studies mainly focus on analyzing the formation mechanism and measurement of marine carbon sinks. Few scholars have examined the supporting role of China’s digital inclusive financial development on marine carbon sink fisheries. Therefore, this paper can provide a reference for scholars to conduct similar studies. In addition, we provide empirical evidence for the study of factors influencing the efficiency of digital financial support. However, there are still some limitations in this paper.

First, the lack of micro-financing data for marine carbon sink fisheries makes it difficult to use the empirical model to confirm that the development of digital finance has alleviated the financing constraints of carbon sink fisheries, which is precisely the direction that should continue to be improved upon in the next study. Second, due to the limitation of data availability, the sample of this paper cannot further support the study of regional heterogeneity in factors influencing the efficiency of digital inclusive financial support. China is a vast geographical area with huge differences in economic development and institutional arrangements, and there are significant differences among regions in the development of digital finance, the strength of environmental regulation, public demand for environmental protection, and the development patterns of mariculture fisheries. Therefore, it is necessary to continue exploring this issue in depth.

## Figures and Tables

**Figure 1 ijerph-19-13971-f001:**
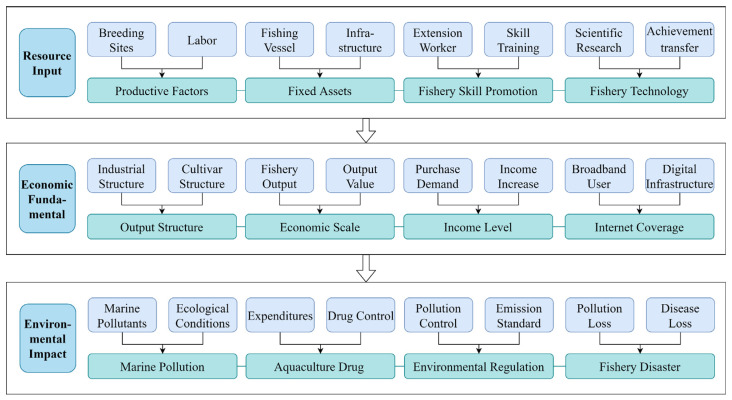
Process diagram of axial coding (level 2 coding).

**Figure 2 ijerph-19-13971-f002:**
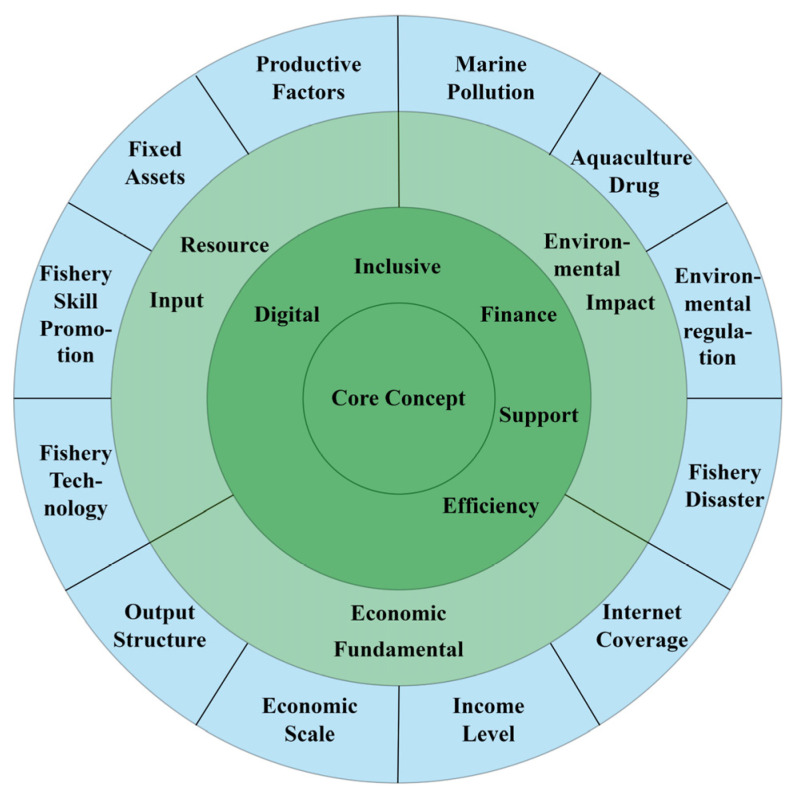
Process diagram of selective coding (level 3 coding).

**Figure 3 ijerph-19-13971-f003:**
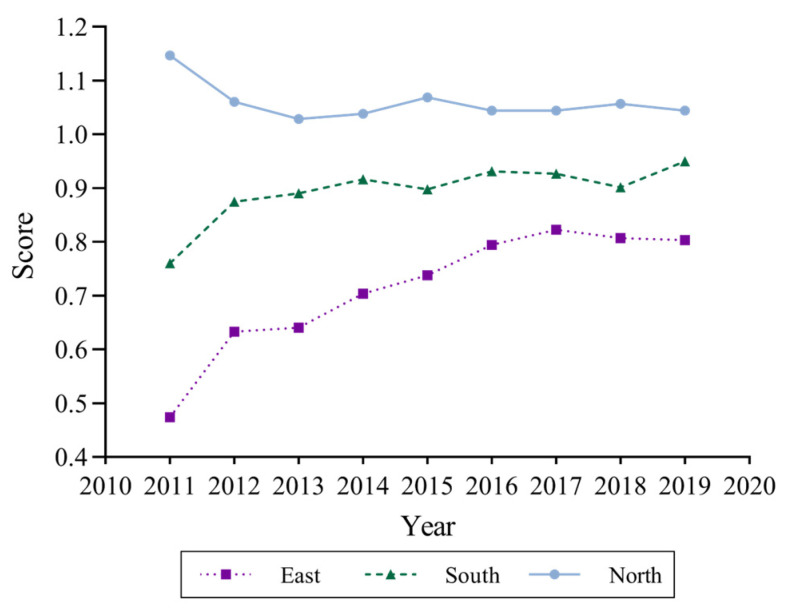
Trends in the DIFSE for marine carbon sink fisheries from 2011 to 2019.

**Figure 4 ijerph-19-13971-f004:**
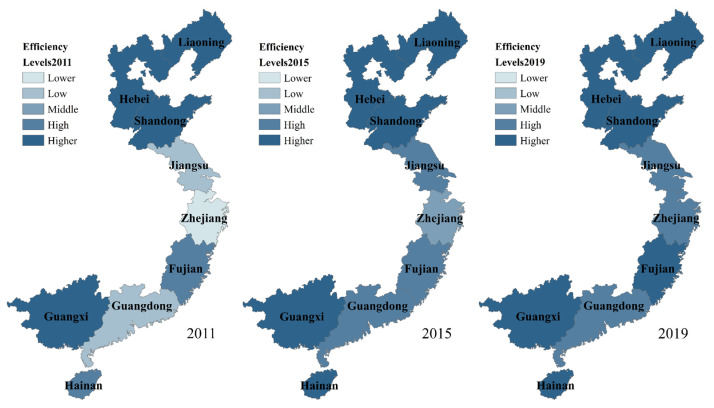
Spatial and temporal distribution of DIFSE for marine carbon sink fisheries.

**Table 1 ijerph-19-13971-t001:** Variable design and descriptive statistics.

Variables	Symbol	Variable Definitions	Mean	Standard Deviation
Digital Inclusive Finance	*DIF*	Digital Financial Inclusion Index (index)	214.34	91.65
Usage Depth	*UD*	Level 1 indicators of *DIF* (index)	215.16	90.63
Coverage Breadth	*CB*	Level 1 indicators of *DIF* (index)	194.81	89.16
Degree of Digitization	*DD*	Level 1 indicators of *DIF* (index)	277.32	119.96
Credit Operations	*CRE*	Level 2 indicators of *DIF* (index)	147.49	53.81
Marine Carbon Sinks	*MCS*	The marine carbon sink capacity (10,000 tons)	13.80	11.65

**Table 2 ijerph-19-13971-t002:** Variable design and descriptive statistics.

	Variables	Symbol	Variable Definitions	Mean	Standard Deviation
Resource input	Productive factors	*factors*	Sum of the standardized values of marine carbon sink fishery breeding sites area, labor, seedlings, power of fishing vessels, and intermediate consumption (index)	1.16	0.67
	Promotion of fishery skill	*skill*	Sum of the standardized values of aquatic skill business promotion funds and the number of fishermen skillfully trained in marine carbon sink fisheries (index)	0.48	0.34
	Fishery technology	*tech*	Sum of the standardized values of the number of research papers and patents related to marine carbon sink fisheries (index)	0.74	0.51
Economic fundamental	Output structure	*structure*	Marine carbon sink fishery production/total mariculture production (ratio)	0.76	0.22
	Income level	*income*	Regional per capita disposable income (yuan)	31,909.37	8339.97
	Internet coverage	*IC*	Regional Internet broadband access users/the number of the regional resident population. (ratio)	0.23	0.09
Environmental impacts	Marine pollution	*pollution*	Marine wastewater discharge (10,000 tons)	69,610.78	70,600.59
	Environmental regulation	*ER*	Number of marine pollution control investment projects (number)	54,161.56	55,156.09
	Fishery disaster	*disaster*	Affected area of breeding sites/total area (ratio)	0.10	0.11
	Aquaculture drug	*drug*	Percentage of fishery drug output value in total mariculture output value (percentage)	0.34	0.74

**Table 3 ijerph-19-13971-t003:** Deng’s grey relational model results in the three marine economic circle.

	Provinces	*DIF*	*CB*	*UD*	*DD*	*CRE*
North	Hebei	0.6568	0.5462	0.7539	0.7063	0.8990
	Liaoning	0.6392	0.6830	0.6533	0.5010	0.7655
	Shandong	0.5706	0.5691	0.6285	0.5028	0.8361
	Mean	0.6222	0.5994	0.6785	0.5700	0.8335
East	Jiangsu	0.8371	0.8659	0.8706	0.5074	0.9283
	Zhejiang	0.8379	0.8767	0.8588	0.5082	0.9318
	Mean	0.8375	0.8713	0.8647	0.5078	0.9301
South	Fujian	0.6458	0.6748	0.6837	0.5039	0.8171
	Guangdong	0.5801	0.5764	0.6316	0.5154	0.7946
	Guangxi	0.6357	0.5385	0.7186	0.6865	0.8175
	Hainan	0.6337	0.5611	0.7115	0.6695	0.8425
	Mean	0.6238	0.5877	0.6863	0.5938	0.8179

**Table 4 ijerph-19-13971-t004:** The spatio-temporal evolution in the DIFSE for marine carbon sink fisheries.

Province	2011	2012	2013	2014	2015	2016	2017	2018	2019	Mean	Rank
Hebei	1.0150	1.0014	1.0028	1.0348	1.0196	1.0146	1.0063	1.0246	1.0057	1.0139	2
Liaoning	1.0378	1.0030	0.9214	0.9471	1.0006	0.9667	0.9862	1.0027	1.0137	0.9866	4
Jiangsu	0.5090	0.6865	0.6897	0.7452	0.7754	0.8247	0.8496	0.8320	0.8290	0.7490	8
Zhejiang	0.4390	0.5796	0.5918	0.6618	0.6999	0.7641	0.7963	0.7818	0.7782	0.6769	9
Fujian	0.7029	0.8513	0.8563	0.8818	0.8376	0.8649	0.8776	0.8756	1.0000	0.8609	6
Shandong	1.3888	1.1782	1.1607	1.1324	1.1869	1.1512	1.1393	1.1426	1.1132	1.1770	1
Guangdong	0.5269	0.7396	0.7409	0.7879	0.8006	0.8549	0.8564	0.8480	0.8403	0.7773	7
Guangxi	1.0557	1.0250	1.0745	0.9771	1.0001	0.9933	1.0009	0.9821	1.0039	1.0125	3
Hainan	0.7562	0.8836	0.8910	1.0172	0.9527	1.0114	0.9735	0.8991	0.9562	0.9268	5
Mean	0.8257	0.8831	0.8810	0.9095	0.9193	0.9384	0.9429	0.9321	0.9489		

**Table 5 ijerph-19-13971-t005:** Tobit regression results.

Variables	Coef.	Std. Err.	t	P
*factors*	0.0971 ***	0.0298	3.2600	0.0020
*skill*	0.1803 *	0.1067	1.6900	0.0950
*tech*	0.0864 ***	0.0243	3.5600	0.0010
*structure*	−0.2204 **	0.0943	−2.3400	0.0220
*Lnincome*	−0.2288 *	0.1257	−1.8200	0.0730
*IC*	0.6593 *	0.3761	1.7500	0.0840
*Lnpollution*	−0.0886 ***	0.0246	−3.6000	0.0010
*LnER*	0.0074	0.0147	0.5100	0.6140
*disaster*	−0.8219 ***	0.2343	−3.5100	0.0010
*drug*	0.0056	0.0347	0.1600	0.8730
Cons	3.9619 ***	1.1476	3.4500	0.0010
Log pseudolikelihood	69.1129		Obs	81

Notes: P statistics in parentheses; *** *p* < 0.01, ** *p* < 0.05, * *p* < 0.1.

## Data Availability

The data presented in this study are available on request from the corresponding author. The data are not publicly available due to future research.

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
