# Peer review of "Evaluation and Determinants of the Digital Inclusive Financial Support Efficiency for Marine Carbon Sink Fisheries: Evidence from China"

_ijerph, 2022, doi:10.3390/ijerph192113971_

Round 1

Reviewer 1 Report

Attachement is the comments and suggestions on the paper.

Author Response

Attachement is the response to reviewer#1.

Reviewer 2 Report

The empirical study developed by the authors using the gray relational analysis model has indisputable scientific value, being at the same time a very good theoretical information support for students and researchers interested in understanding and applying this reliable method.

The manuscript is developed with professionalism and is based on experimental data collected over an extended period of time, from several coastal regions of China. The text is properly structured, being supported by eloquent graphic images, of good quality.

Introduction and the literature study are rich in relevant information, the authors citing fairly current articles. The novelty and originality of this study are revealed and the aim is well defined. Research methodology is described in detail and the models used are presented accordingly.

The results are exposed coherently and are analyzed in depth. I note that the authors also brought to light the limitations of the model used and pointed out the issues that should be further investigated in future studies.

I have two recommendations that may improve the quality of this manuscript:

1. Many abbreviations are not defined in full (e.g. lines 144, 186, 201, 211, 315, 348), I recommend defining them or including a table of abbreviations.

2. Conclusions section is missing; I suggest the authors to include this section in a concentrated format, which summarizes the main results and directions for further study.

Author Response

Attachement is the response to reviewer#2.

Reviewer 3 Report

Dear Authors, the article is very interesting. However, I am including some recommendations to improve it.

1) Introduction: Lines 28-47: Please, add another reference. As matter of fact, in Scopus there are 139 products elaborated with the strings "digital AND inclusive AND finance".

2) Introduction: Lines 78-81: plesse, insert a novelty of the study.

3) Literature review: line 118: 6.5 Pg (C) of CO2 annually: what'is Pg of CO2?

4) Manuscript: It is necessary to explain the units of measurement by giving a definition of the indicator you are using because they are too many and it can be confusing.

5) Lines 221-223: "Grounded theory is a qualitative research method based on empirical data to form the concepts, categories, and connections of the research object, and ultimately establish a theory from the bottom up [34]. This sentence is very interesting.

6) Page 5, line 239. In this section, please insert a figure of level 1 of coding, as already done for level 2 and 3 of coding.

7) Tables 1-2: Please reduce the size of these tables.

8) Tables 3-4-5: Plaese, create a single table that summarizes the data.

9) Software for regression is Stata 17.0. Please, indicate and describe this in the methodology, not only in the results.

10) Page 20: lie 787: In my oponion is necessary insert the conclusions, limitations and future implications paragraph in order to lead the scholar to a closing evaluation, detached from the discussions.

Author Response

Attachement is the response to reviewer#3.

Round 2

Reviewer 1 Report

This revised manuscript has been improved sufficiently.